# Facultative CTCF sites moderate mammary super-enhancer activity and regulate juxtaposed gene in non-mammary cells

M. Willi[1,2,*], K.H. Yoo[1,3,*], F. Reinisch[1], T.M. Kuhns[1], H.K. Lee[1,4], C. Wang[1] & L. Hennighausen[1]

Precise spatiotemporal gene regulation is paramount for the establishment and maintenance of cell-specific programmes. Although there is evidence that chromatin neighbourhoods, formed by the zinc-finger protein CTCF, can sequester enhancers and their target genes, there is limited *in vivo* evidence for CTCF demarcating super-enhancers and preventing cross talk between distinct regulatory elements. Here, we address these questions in the *Wap* locus with its mammary-specific super-enhancer separated by CTCF sites from widely expressed genes. Mutational analysis demonstrates that the *Wap* super-enhancer controls *Ramp3*, despite three separating CTCF sites. Their deletion in mice results in elevated expression of *Ramp3* in mammary tissue through augmented promoter–enhancer interactions. Deletion of the distal CTCF-binding site results in loss of *Ramp3* expression in non-mammary tissues. This suggests that CTCF sites are porous borders, allowing a super-enhancer to activate a secondary target. Likewise, CTCF sites shield a widely expressed gene from suppressive influences of a silent locus.

[1] Laboratory of Genetics and Physiology, National Institute of Diabetes and Digestive and Kidney Diseases, US National Institutes of Health, Bethesda, Maryland 20892, USA. [2] Division of Bioinformatics, Biocenter, Medical University of Innsbruck, 6020 Innsbruck, Austria. [3] Department of Life Systems, Sookmyung Women's University, Seoul 140-742, Republic of Korea. [4] Department of Cell and Developmental Biology & Dental Research Institute, Seoul National University, Seoul 110-749, Republic of Korea. * These authors contributed equally to this work. Correspondence and requests for materials should be addressed to L.H. (email: lotharh@mail.nih.gov).

Super-enhancers control lineage-specific genes[1–5] and activate them up to several thousand-fold[6]. In order to prevent a spill over of transcriptional activity, such loci need to be self-contained units and demarcated from neighbouring genes that are subject to their own distinct control. This ensures that neighbouring non-target genes are correctly regulated. Equally important, active genetic units need to be impervious to flanking heterochromatin to avoid inappropriately silencing. Experimental evidence suggests that genes and their associated regulatory elements are located within genetically confined neighbourhoods[7–12]. The concept of a genome compartmentalized into regulatory neighbourhoods[7–12], such as topological-associated domains (TADs)[9,11] and insulated neighbourhoods[7,8,12], was established along with the identification of chromatin loops using Chromosome Conformation Capture (3C) technologies, Chromatin Interaction Analysis by Paired-End Tag sequencing (ChIA-PET)[13] and Hi-C[14], and the accompanying computational approaches. Such regulatory neighbourhoods not only limit the search space of cis-regulatory elements to genes within the same domain but also prevent active and suppressive chromatin from spreading[7,15–18]. The zinc-finger protein CTCF, which is known for its insulator function[19–21], has been shown to be instrumental in establishing such insulated neighbourhoods[7] and is also enriched at TAD borders[9]. High-resolution data suggest a size of approximately 185 kb for TADs[22], called contact domains[22]. The additional presence of sub-domain chromatin loops within TADs enables accurate promoter–enhancer interactions necessary for the developmental control of resident genes[22].

Current knowledge about biological functions of CTCF sites has been obtained mainly from studies on cell lines and embryonic stem cells (ESCs)[7,10,15,16,18,23–26]. Only a limited number of mouse studies have been conducted[17,25,27–31] and these have in general not addressed the extent to which CTCF sites can shield non-target genes from juxtaposed enhancers. Naturally occurring human mutations and engineered mice carrying the mutations in CTCF sites have provided additional information on their contribution in developmental diseases and tumorigenesis[17,26,31–36]. However, there has been no genetic assessment of the contribution of CTCF sites in confining cell-specific and cytokine-sensing enhancers in a genuine in vivo setting. Cytokine-sensing super-enhancers can activate genes up to several thousand-fold and it is not clear whether CTCF sites can demarcate them to avoid aberrant regulation of outside genes.

In a quest to understand the physiological role of CTCF in the cell-specific control of super-enhancers, we turned to the mammary gland, whose sole purpose is the production of milk to nourish the young. More than 90% of the protein content of milk is contributed by less than 10 proteins[37] and expression of the respective genes in alveolar epithelium is induced up to 1,000-fold during pregnancy[38]. The extraordinary expression levels of milk protein genes are the result of mammary-specific super-enhancers that integrate prolactin signalling during pregnancy and lactation[6]. While highly active in mammary alveolar epithelium, most milk protein encoding genes are virtually silent in non-mammary cells, even in those that sense cytokines including prolactin. Mammary-specific genes are surrounded by genes expressed across cell types and are subject to their own regulatory features. It is essential that mammary super-enhancers only activate their respective target genes and neighbouring off-target genes remain inert to their influence. Inadvertent and exorbitant expression of any biologically active protein in mammary tissue could be detrimental to its physiology[39]. Since mammary enhancers are composed of common elements that can be activated by many cytokines, it is essential to shield their influence from neighbouring genes in non-mammary cells. Equally important, it is vital that the transcriptional silence of mammary loci in non-mammary cells does not spread to active neighbouring genes, which could result in their inadvertent silencing with all its potential consequences. At this point it is not known whether CTCF sites demarcate highly active mammary super-enhancers from neighbouring genes.

Here, we have used the well-characterized Wap locus to investigate the biological role of CTCF sites controlling mammary-specific loci. The Wap gene itself is expressed exclusively in mammary epithelium and its more than 1,000-fold activation during pregnancy is controlled by a tripartite super-enhancer that senses prolactin through the transcription factor STAT5 and likely other mammary-enriched transcription factors[6]. The Wap locus is compact and Ramp3, which is expressed in many cell types at low levels, is located within 14 kb of the super-enhancer. Tbrg4, which is also expressed at low levels in many cell types, flanks the other side of the Wap locus. Both RAMP3 (ref. 40) and TBRG4[41] are regulatory proteins and well-defined levels likely ensure biological functions of expressing cells.

We identify five CTCF-binding sites in mammary tissue that separate the Wap gene with its super-enhancer from its neighbouring genes using ChIP-seq. Based on these findings we address four specific questions. Is the Wap gene located within a unique regulatory domain separated from neighbouring genes? Does the Wap super-enhancer activate the two neighbouring non-target genes in mammary tissue? Do CTCF sites shield the Wap super-enhancer from neighbouring genes in mammary tissue? Do CTCF sites shield the silent Wap gene in non-mammary cells from common regulatory elements controlling the two neighbouring genes?

We demonstrate that CTCF sites do not constitute a tight genetic boundary encapsulating the mammary-specific Wap super-enhancer and its associated gene, but are a porous border that tolerates enhancer spill over to the neighbouring non-target Ramp3 gene. We also demonstrate that a conserved CTCF site separating the Wap super-enhancer from the Ramp3 gene is required for Ramp3 expression in non-mammary tissues. We show that CTCF-binding sites contribute to cell-specific functions in the activation and silencing of genes.

## Results

**Identification of CTCF sites associated with super-enhancers.** Mammary super enhancers activate their respective target genes up to 1,000-fold during pregnancy but have little or no impact on neighbouring non-target genes[6]. To investigate the possibility that the zinc-finger protein CTCF contributes to shielding mammary enhancers from non-target genes, we initially analysed genome-wide binding of CTCF in mammary tissue at day one of lactation (Fig. 1). ChIP-seq experiments demonstrated the presence of approximately 26,200 CTCF-binding sites. Based on a comparison with published ChIP-seq data from other tissues, 98% of these binding sites were shared with those found in one or more tissues (Th17 cells, kidney, heart, cerebellum and adipose tissue) and 2% (588) appeared to be mammary-specific. CTCF-binding sites shared between tissues were distinct from mammary-specific sites (Fig. 1b). Shared sites had on average a 1.6-fold higher read coverage than mammary-specific sites (Fig. 1b). While 74% of the common peaks displayed a bona fide binding motif, so did 57% of the mammary-specific ones (Fig. 1c).

Next, we investigated the possibility that mammary super-enhancers were associated with specific CTCF sites. Ninety-six per cent of the 370 CTCF sites located within the 440 mammary super-enhancers[6] were shared with several cell types (Fig. 1d) and their read coverage exceeded that of mammary-specific sites by

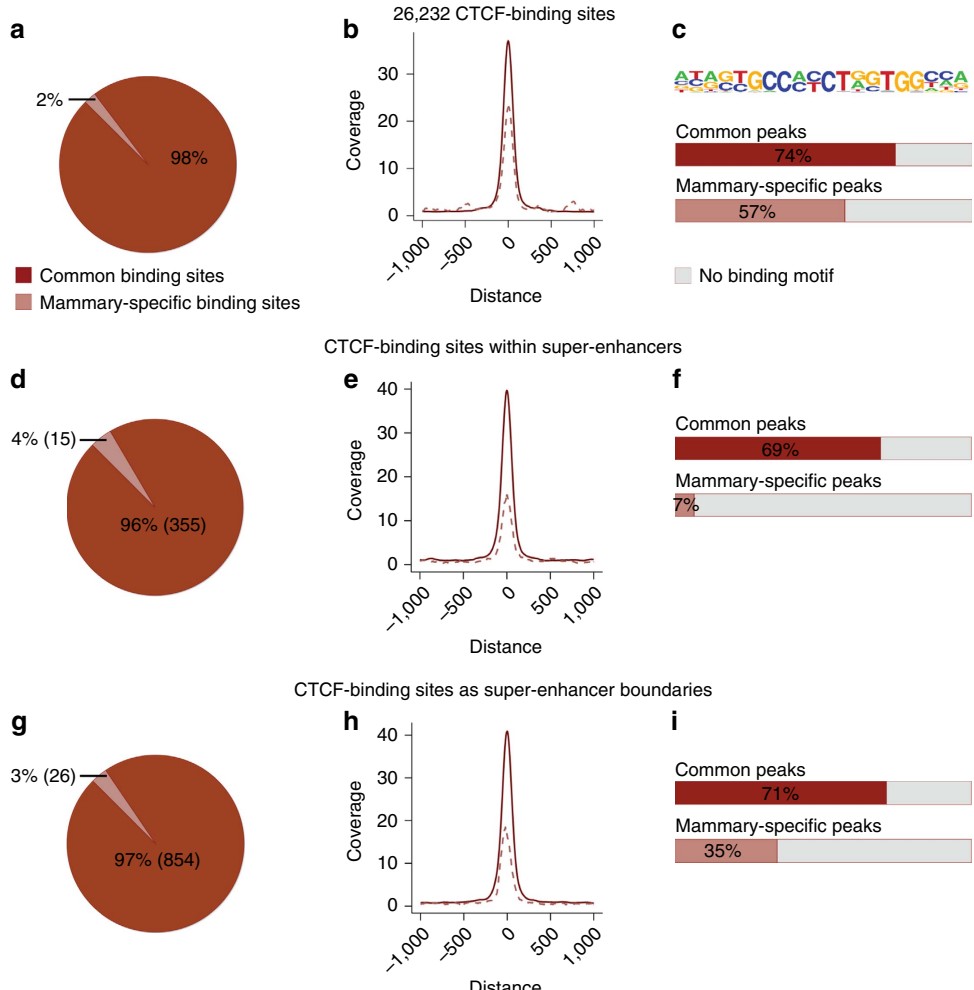

**Figure 1 | Global analysis of CTCF-binding sites.** A total of 26,232 CTCF-binding sites were identified by ChIP-seq of mammary tissue from lactating mice, by peak calling and overlapping three CTCF ChIP-seq samples (**a**) Ninety-eight per cent (25,644) of the sites were shared between mammary tissue, adipose tissue, cerebellum, heart, kidney and Th17 cells (GSE29218, GSE40918, GSE74189, GSE74826). Out of those, 65% (16,733) have been identified in all tissues and 35% (8,911) in distinct subsets. Two per cent (588) of the sites appeared to be mammary-specific. (**b**) Read coverage of CTCF-binding sites shared between tissues (height 37) was approximately 1.6 times higher than the mammary-specific ones (height 23). (**c**) The CCCTC motif was the highest ranked motif in both peak sets ( ± 200 bp). Seventy-four per cent (19,057) of CTCF-binding sites shared between tissues had an underlying CCCTC-binding motif and 57% (337) of the mammary-specific binding sites had one. (**d**) A total of 370 CTCF-binding sites were detected within the 440 mammary super-enhancers. Ninety-six per cent (355) of them were shared CTCF-binding sites and 4% (15) were mammary-specific. (**e**) The read coverage of CTCF-binding sites shared between tissues (height 40) was approximately 2.5 times higher than those of mammary-specific ones (height 16). (**f**) Sixty-nine per cent (246) of the shared CTCF-binding sites and one site (7%) of the mammary-specific ones had an underlying CTCF motif. (**g**) CTCF-binding sites associated with 440 mammary super-enhancer boundaries. Ninety-seven per cent (854) were shared between tissues and 3% (26) were mammary-specific. (**h**) Read coverage of CTCF-binding sites shared between tissues (height 41) was approximately two times higher than that of mammary-specific ones (height 19). (**i**) Seventy-one per cent (610) of shared CTCF-binding sites, and 35% (nine) of mammary-specific ones had an underlying CTCF motif.

2.5-fold (Fig. 1e). Moreover, while 69% of the shared sites had a CTCF recognition motif, only 7% of the mammary-specific ones displayed one (Fig. 1f). Similarly, 97% of the 854 CTCF sites flanking mammary super-enhancers were shared with other tissues (Fig. 1g), their peak coverage was higher than that of mammary-specific ones (Fig. 1h) and 71% of those shared binding sites had a CTCF motif (Fig. 1i). In summary, most CTCF-binding sites associated with mammary super-enhancers are not unique to mammary tissue.

**The *Wap* super-enhancer activates the juxtaposed *Ramp3* gene.** We have recently identified and dissected a complex mammary super-enhancer that activates the *Wap* gene more than 1,000-fold during pregnancy[6]. The extended locus contains the *Wap* gene,

whose expression is confined to mammary epithelium, and two genes, *Ramp3* and *Tbrg4*, which are active across cell types. This locus contains at least five CTCF sites that are shared between different cell types (Fig. 2a). CTCF binding to site A, which coincides with a TAD boundary[9], and sites E and F, which presumably serve as loop anchors[7,22], were conserved across cell types (Fig. 2a,b). CTCF binding to sites C and D was preferentially found in cells that respond to cytokines, such as T cells.

Based on Hi-C[22] and ChIA-PET[7] data sets, *Ramp3* is located within its own distinct chromatin loop separated from the highly active *Wap* gene (Fig. 2c). Notably, H3K27ac marks covering the *Wap* super-enhancer did not extend past bordering CTCF sites into the *Ramp3* locus (Fig. 2c). However, these structural data provided no information on the extent to which *Wap* super-

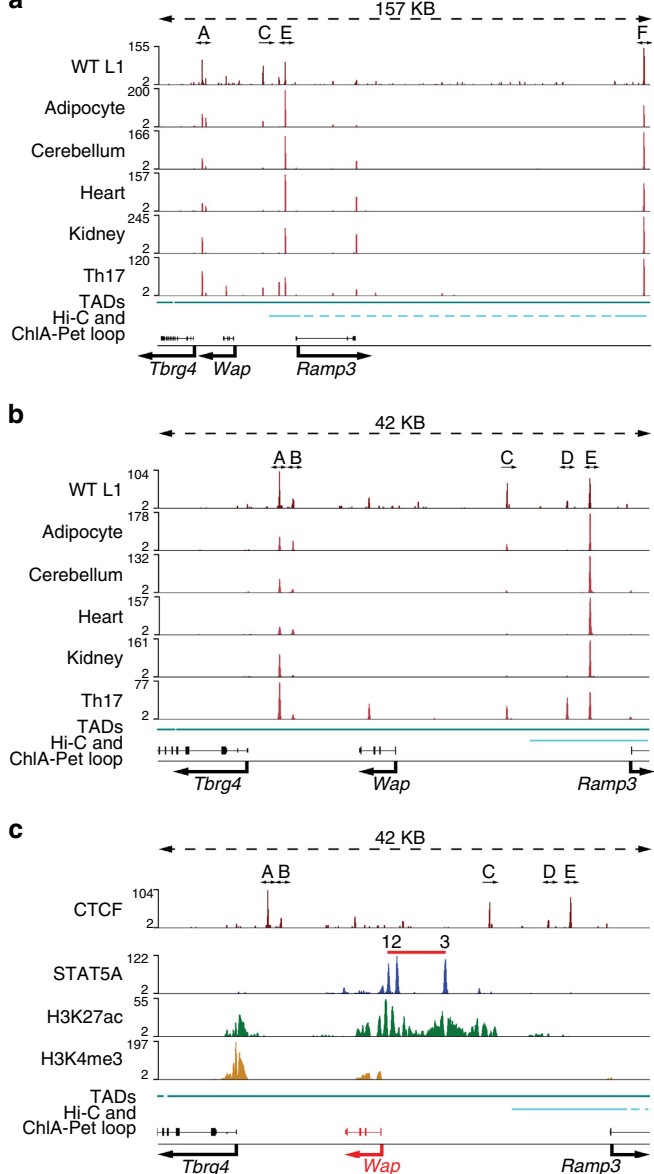

**Figure 2 | CTCF-binding sites and enhancers in the extended *Wap* locus.**
(**a**) CTCF ChIP-seq data from mammary tissue at day 1 of lactation and from several non-mammary cells (GSE29218, GSE40918, GSE74189, GSE74826). The lower part shows the topological associated domain[9] (TAD) (bin size 40KB) in cyan and a ChIA-PET and Hi-C loop[7] with site E and F as anchor in light blue. (**b**) CTCF binding to sites A and F was conserved across cell types. CTCF binding to sites B, C and D was cell-preferential. (**c**) The *Wap* locus is characterized by its tripartite mammary-specific super-enhancer (shown as red bar) and five flanking CTCF sites (GSE74826), which separate it from the two neighbouring genes, *Tbrg4* and *Ramp3*. Both are expressed at low levels (FPKM < 13) in mammary tissue. *Ramp3* is preferentially expressed in the cerebellum, whereas *Tbrg4* is widely expressed. Two CTCF-binding sites are 3′ of the *Wap* gene body and three 5′ sites are located between the super-enhancer and *Ramp3*. The *Wap* locus is located within a topological-associated domain[9] (TAD) (bin size 40KB). CTCF site E anchors a ChIA-PET and Hi-C loop[7].

enhancer elements could activate the neighbouring *Ramp3* gene. RNA-seq experiments[38] demonstrated that while *Wap* was induced 1,000-fold between day 6 of pregnancy and day 1 of lactation, *Ramp3* mRNA levels increased approximately five-fold (Fig. 3a). In contrast, *Tbrg4* mRNA levels remained constant. This

suggests the possibility that *Ramp3* is controlled to some extent by the juxtaposed *Wap* super-enhancer. Alternatively, the increase of *Ramp3* mRNA levels during pregnancy could merely reflect an increased proportion of secreting epithelial cells, which might preferentially express *Ramp3*.

We investigated the possibility that *Ramp3* is under control of the juxtaposed *Wap* super-enhancer, which is composed of three constituent enhancers (Fig. 3b), with the most distal S3 accounting for 90% of its activity[6]. Deletion of S3 (ΔS3) resulted in an approximately 70% reduction of *Ramp3* mRNA in mammary tissue (Fig. 3c). Although *Wap* and *Ramp3* are expressed at vastly different levels in mammary tissue, the relative impact of the super-enhancer was equivalent on both genes despite the presence of three CTCF sites separating it from *Ramp3*. Since wild type and mutant mammary tissues were obtained from lactating mice, we can exclude the possibility that non-mammary cells were responsible for the differential *Ramp3* expression between pregnancy and lactation. While the *Wap* super-enhancer and its associated regulatory regions are characterized by extensive H3K27ac, few, if any, of these marks were detected over the *Ramp3* gene (Fig. 3d). H3K27ac coverage of the *Wap* region was greatly reduced in ΔS3 mammary tissue and absent at the *Ramp3* gene[6] (Fig. 3d). The *Ramp3* gene also displayed a paucity of H3K27ac marks in Th17 cells and adipose tissue (Fig. 3e) reflecting low expression levels. The *Il2rα* gene served as a positive H3K27ac control in Th17 cells (Fig. 3f) and *Mir193b* (ref. 42) in adipocytes (Fig. 3g). *Tbrg4* mRNA levels were not significantly reduced in ΔS3 mutant mammary tissue, suggesting spatial selectivity of the S3 enhancer. Of note, the vastly higher expression of *Wap* compared to *Ramp3* might be caused by additional regulatory elements or differential stability of the two mRNAs.

**CTCF sites moderate super-enhancer activity.** Although the S3 enhancer of the *Wap* super-enhancer activates *Ramp3* despite the presence of three separating CTCF sites, it is not clear to what extent these sites muffle super-enhancer activity. To investigate this, we deleted these CTCF sites individually and in combination from the mouse genome (Fig. 4). While CTCF binding to site E, an anchor of the loop that encapsulates *Ramp3*, is conserved across all cell types analysed (Fig. 2a), CTCF binding to sites C and D is also found in a subset of cells, such as T cells. In addition to these three sites we also investigated the two CTCF-binding sites demarcating the *Wap* gene from its downstream neighbour *Tbrg4* (Fig. 4a). Deletion of these two sites in the mouse germline resulted in the loss of CTCF binding (Fig. 4b) but did not result in an altered expression of *Tbrg4* or *Wap* in lactating mammary tissue (Fig. 4c). This suggests that the *Wap* super-enhancer is unable to regulate *Tbrg4*, which is in accordance with studies in other systems[25]. Alternatively, a weak CTCF-binding site located in the third intron of the *Wap* gene could function as a boundary element.

We asked whether the three CTCF sites separating the *Wap* super-enhancer from *Ramp3* have any inherent insulating capacity and deleted them in mice using CRISPR/Cas9 gene editing (Fig. 4a). Loss of CTCF binding in the different mutants was confirmed by ChIP-seq experiments (Fig. 4b). Next, we investigated the impact of individual deletions on the expression of *Wap* and *Ramp3* in mammary tissue (Fig. 4d). None of the individual or combined deletions impacted *Wap* expression at the onset of lactation (Fig. 4d). However, a differential impact on the neighbouring *Ramp3* gene was obtained (Fig. 4d). While loss of sites C or D was inconsequential for *Ramp3* expression, loss of site E resulted in an approximately five-fold induction of *Ramp3* (Fig. 4d). Even though site E seems to be the only one to

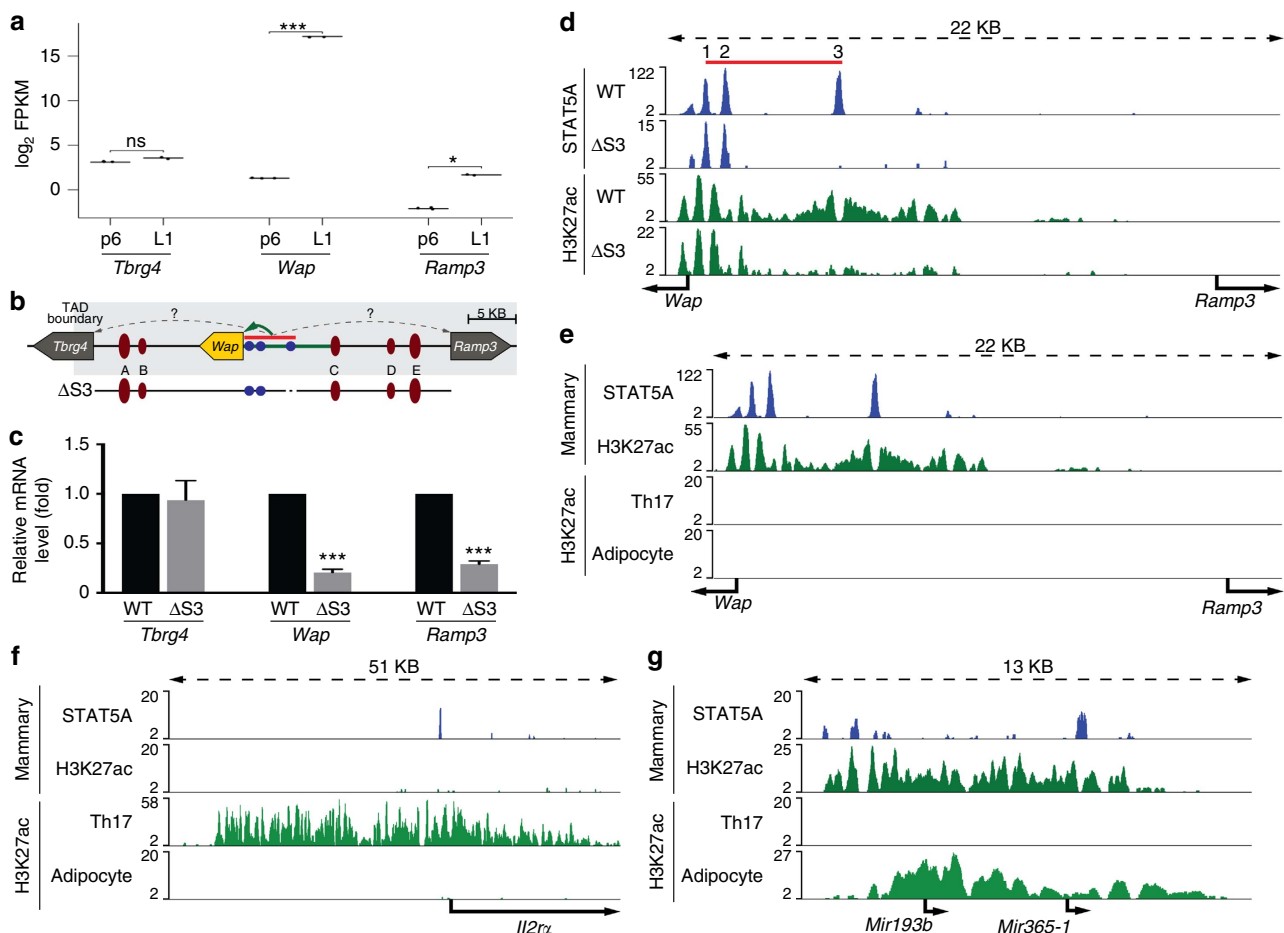

**Figure 3 | Wap super-enhancer activates the Ramp3 gene.** (**a**) RNA-seq data from mammary tissue at day 6 of pregnancy (p6) and day 1 of lactation (L1) demonstrated that Wap and Ramp3 are induced during pregnancy, although Ramp3 is expressed at a much lower level. Tbrg4 expression remained unchanged (GSE37646). Results are shown as box plots; median, middle bar inside the box; IQR, 50% of the data; whiskers, 1.5 times the IQR; p6: $n = 3$ and L1: $n = 2$. Student's $t$-test with alpha 0.01 was applied for p6 and L1 for each gene independently. $*P < 0.01$, $**P < 0.001$, $***P < 0.0001$. (**b**) The cartoon illustrates Wap and its flanking genes. The blue circles represent the constituent enhancers of the super-enhancer (shown as red bar). H3K27ac is shown in green. The crimson ovals represent the sites bound by CTCF. (**c**) Gene expression data from the enhancer mutant ΔS3 demonstrated that Ramp3 expression was reduced by 70%. Results are shown as the means ± s.e.m. of independent biological replicates; ΔS3: $n = 3$; data were normalized to Gapdh; a two-tailed Student's $t$-test was applied with alpha 0.01 for wild type and ΔS3 for each gene independently. $*P < 0.01$, $**P < 0.001$, $***P < 0.0001$. (**d**) STAT5 binding and H3K27ac in the Wap-Ramp3 locus in WT and ΔS3 mutant mammary tissue (GSE74826). (**e**) H3K27ac at the Wap-Ramp3 locus in mammary tissue, Th17 cells and adipocytes (GSE90788, GSE92590). (**f**) H3K27ac at the Il2rα locus in mammary tissue, Th17 cells and adipocytes. (**g**) H3K27ac at the Mir193b locus in mammary tissue, Th17 cells and adipocytes.

individually muffle Wap super-enhancer activity, we also investigated the consequences upon loss of all three sites. The combined loss of these sites resulted in an approximately seven-fold increase of Ramp3 mRNA (Fig. 4d). This suggests that sites C and D have measurable biological activity only in the context of site E. Although milk-secreting cells constitute the vast majority of mammary epithelium during lactation, it is possible that Ramp3 is expressed at significant levels in other cell types. To account for this, we analysed Wap and Ramp3 mRNA levels in mammary tissue from non-parous mice (Fig. 4e). The combined loss of CTCF sites C, D and E did not significantly alter the expression of Wap and Ramp3 genes in mammary tissue from virgin mice, suggesting that the transcriptional activation of the Ramp3 gene during pregnancy is the result of Wap enhancer activity.

We further examined the possibility that this induction was the result of an expansion of Wap super-enhancer structures. Notably, the combined loss of sites C, D and E resulted in elevated H3K27ac over the Ramp3 gene (Fig. 5a). De novo

H3K27ac acquisition was preferentially obtained in the first intron of Ramp3 and coincided with increased CTCF binding in these mutants (Fig. 6b). We also observed an expansion of H3K27ac in mammary tissue lacking only site C, which directly borders the Wap super-enhancer. However, this enhanced H3K27ac, which extended to site E, was without measurable consequences on gene expression (Fig. 4d). The H3K27ac pattern at the Bcl6 locus served as control accommodating variations between samples (Fig. 5b). Densitometry of H3K27ac occupancy supported the expansion of enhancer marks into the Ramp3 locus (Fig. 5c,d).

RNAs associated with enhancers (eRNAs) tend to reflect their activities[43]. To determine whether increased expression of Ramp3 in mammary tissue lacking CTCF site E was caused by altered Wap super-enhancer activity, we examined eRNA levels. Total RNA-seq established the eRNA pattern over S3, the predominant constituent enhancer within the super-enhancer (Fig. 7). Transcripts coincided with S3 and two additional STAT5-binding sites, which were also associated with H3K27ac marks

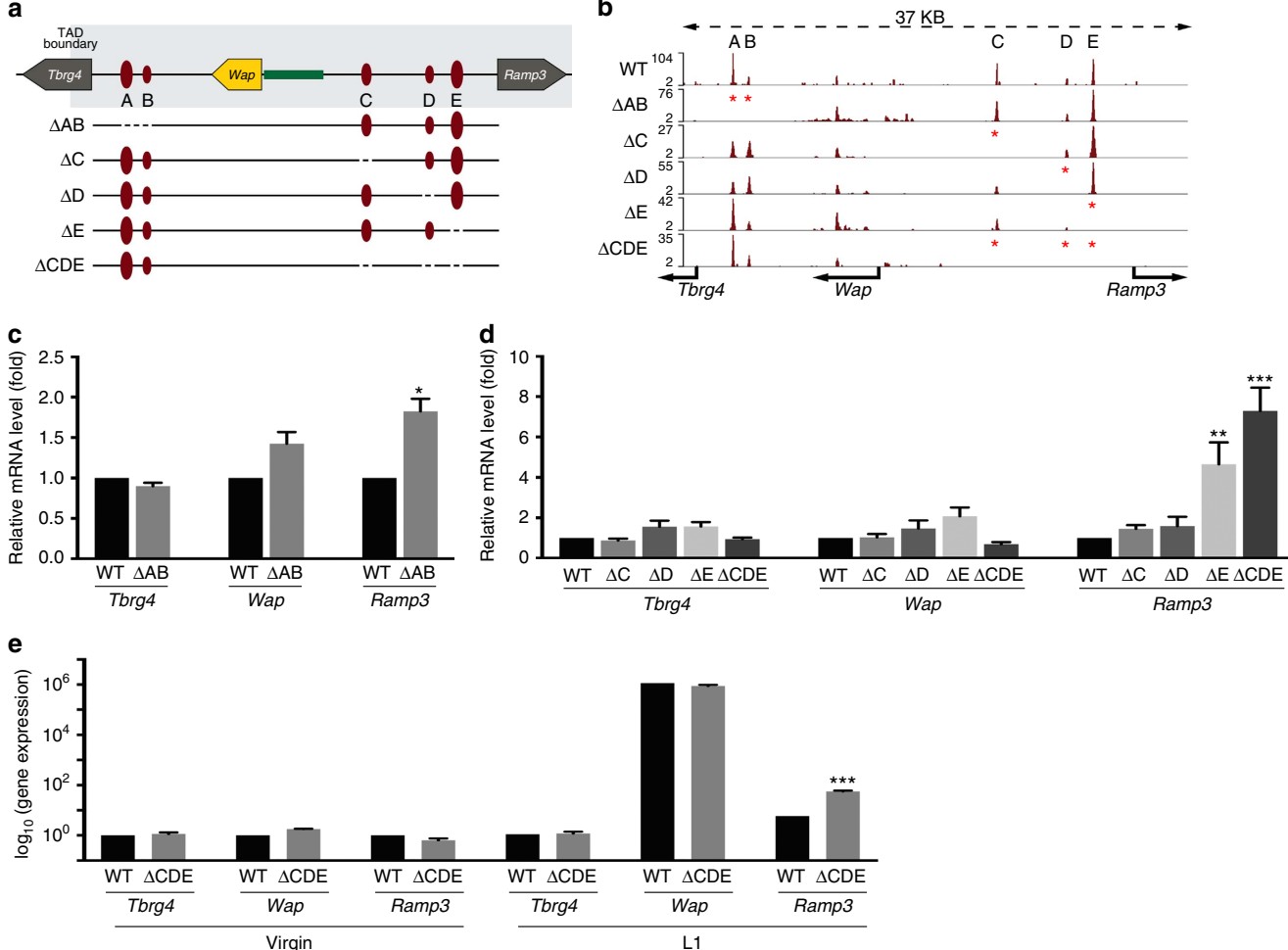

**Figure 4 | Deletion of CTCF sites flanking the *Wap* locus. (a)** Diagram of the CTCF-binding sites in the *Tbrg4-Wap-Ramp3* locus and the deletions investigated. Individual deletions of sites C, D and E and two combined deletions, one comprising sites A and B and one comprising sites C, D and E were generated and analysed. **(b)** CTCF ChIP-seq analyses from mutant tissues confirmed loss of CTCF binding at the respective deleted sites. **(c)** Deletion of sites A and B did not affect expression of *Tbrg4* and *Wap* using qRT–PCR. Results are shown as the means ± s.e.m. of independent biological replicates; ΔAB $n = 4$ ( − / − ); data were normalized to *Gapdh*; a two-tailed Student's *t*-test was applied with alpha 0.01 for wild type and ΔAB for each gene independently. *$P < 0.01$, **$P < 0.001$, ***$P < 0.0001$. **(d)** *Tbrg4*, *Wap* and *Ramp3* mRNA levels in lactating mammary tissue lacking CTCF sites separating these two genes. Loss of sites C or D did not affect gene expression, whereas loss of site E resulted in an approximately five-fold increase in expression of *Ramp3* and a seven-fold induction in mice carrying combined deletions of C, D and E. *Tbrg4* and *Wap* did not show changes in gene expression. Results are shown as the means ± s.e.m. of independent biological replicates; ΔC $n = 10$ ( − / − ); ΔD $n = 5$ ( − / − ); ΔE $n = 7$ ( − / − ); ΔCDE: $n = 4$ ( − / − ); data were normalized to *Gapdh*; one-way ANOVA with multiple comparison and alpha 0.01 was applied. *$P < 0.01$, **$P < 0.001$, ***$P < 0.0001$; **(e)** *Tbrg4*, *Wap* and *Ramp3* mRNA levels in virgin mammary tissue lacking the three CTCF sites separating the *Wap* super-enhancer from the *Ramp3* gene. A two-tailed Student's *t*-test was applied with alpha 0.01 for wild type and ΔCDE for each gene independently. *$P < 0.01$, **$P < 0.001$, ***$P < 0.0001$.

as well as Pol II binding. Using 5′ and 3′ RACE we determined the exact structure of the eRNA emanating from S3 (Fig. 7a, Supplementary Note 2). S3 eRNA levels were determined by qRT–PCR and no significant differences were observed between wild type and the three mutant tissues (Fig. 7b). This suggests that loss of CTCF sites flanking the super-enhancer does not affect its overall activity but rather extends its sphere of influence.

**Elevated *Wap* enhancer interactions upon loss of CTCF sites.** To understand the mechanism underlying the activation of *Ramp3* in mammary tissue, especially upon loss of the three CTCF-binding sites separating it from the *Wap* super-enhancer, we conducted 4C experiments. Using the *Wap* super-enhancer as anchor we determined the interactions inherent to the native locus and elevated or *de novo* interactions obtained upon loss of CTCF sites (Fig. 6). We investigated whether the *Wap* super-

enhancer interacts with the *Ramp3* gene and 4C experiments in wild type tissue demonstrated interactions of the S3 region with the *Ramp3* gene (Fig. 6a). In accordance with our previous findings this interaction was abrogated in ΔS3 mutant mammary tissue. Upon deletion of all CTCF sites separating the *Wap* super-enhancer from *Ramp3*, elevated interactions between the S3 region and the first intron of *Ramp3* were detected (Fig. 6b, upper panel). The underlying region coincided with augmented H3K27ac coverage (Figs 5a and 6b, bottom panel) and increased CTCF binding in ΔCDE mutants (Fig. 6b, centre panel).

**CTCF sites ensure *Ramp3* expression in non-mammary tissues.** A characterizing feature of the *Wap* locus is its absolute silence in non-mammary tissues, which, however, soundly express the neighbouring *Ramp3*. The presence of H3K4me3 marks on the *Ramp3* promoter in non-mammary cells and tissues (Fig. 8a)

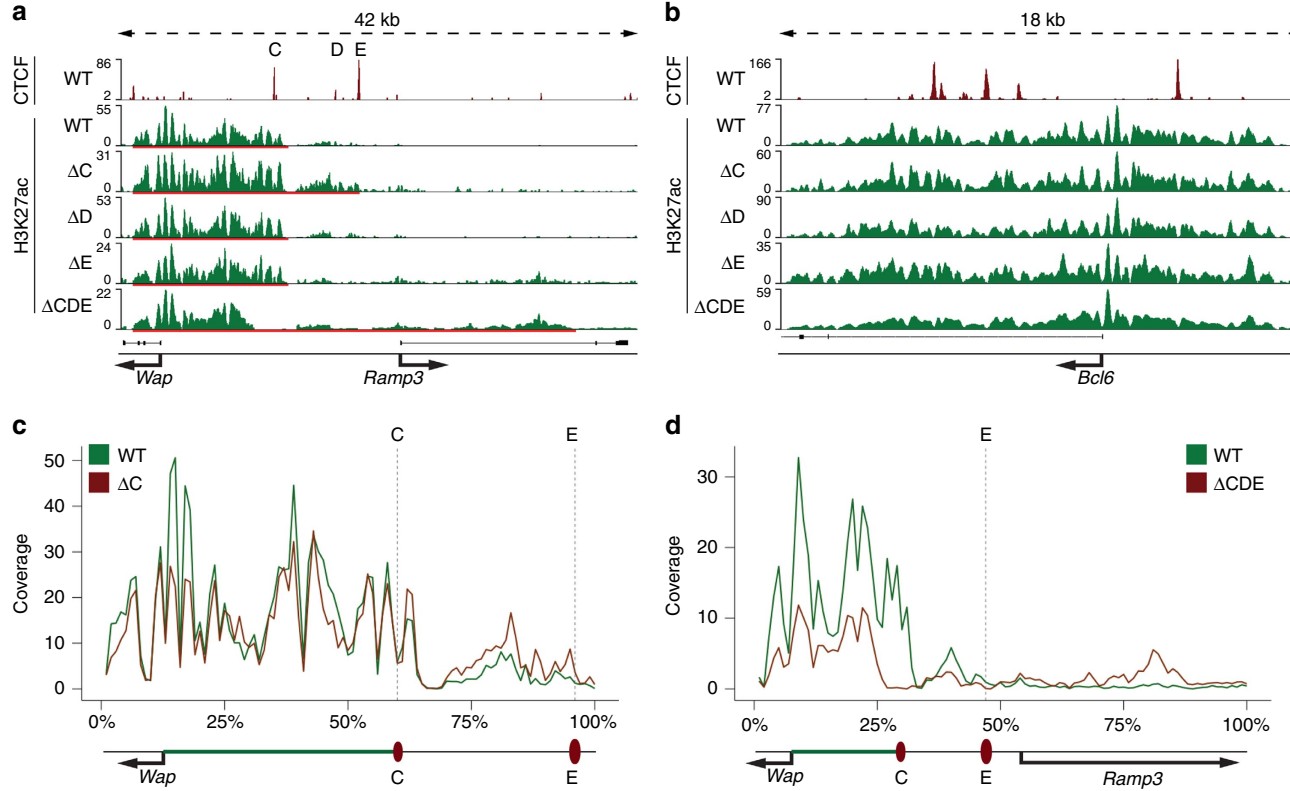

**Figure 5 | Expansion of H3K27ac marks upon deletion of CTCF sites.** ChIP-seq was used to identify changes of H3K27ac due to mutated CTCF-binding sites. (**a**) Deletion of CTCF site C resulted in the expansion of H3K27ac associated with the *Wap* super-enhancer (red bar). This was not observed upon loss of sites D or E. The combined loss of sites C, D and E resulted also in an expansion of H3K27ac and was particular strong at a specific site in the first intron of *Ramp3* (normalized to 10 million reads) (red bar). (**b**) H3K27ac intensity was equivalent in the unrelated *Bcl6* locus. (**c**) H3K27ac profiles of the *Wap-Ramp3* locus based on the data shown in **a**. Green, WT pattern; red, ΔC. The degree of H3K27ac in mutants was elevated between CTCF sites C and E confirming the expansion of H3K27ac (normalized to 10 million reads). (**d**) H3K27ac profiles of the data shown in **a**. WT is shown in green, ΔCDE in red. H3K27ac marks spread from site E to the second exon of *Ramp3* (normalized to 10 million reads).

provides evidence of its activity. This opened the possibility of the three CTCF sites shielding *Ramp3* regulatory elements and thereby ensuring the silence of *Wap* in non-mammary cells. We initially addressed this question in the cerebellum, a tissue characterized by high *Ramp3* expression and the absence of any detectable *Wap* mRNA. ChIP-seq analysis had demonstrated that from the three CTCF sites only site E was bound by CTCF in the cerebellum (Fig. 2b). Individual absence of CTCF sites D or E, nor the combined absence of sites C, D and E, failed to activate *Wap* (Fig. 8b,c). This demonstrates an inability of *Ramp3* regulatory elements to spread into the *Wap* locus in non-mammary cells. Notably, deletion of CTCF site E resulted in a 90% reduction of *Ramp3* mRNA levels in the cerebellum (Fig. 8c). The combined deletion of all three sites also led to a 90% reduction of *Ramp3*. Similarly, reduced expression was also observed in other non-mammary tissues (Fig. 8d), including kidney and uterus. Finally, we investigated the possibility that CTCF site E coincides with putative regulatory elements. ChIP-seq data demonstrated the presence of strong DNase I hypersensitivity coinciding with CTCF site E and non-mammary cells (Fig. 8e).

In some cell types the presence of H3K27me3 marks has been linked to gene inactivity[44,45] and along with that, CTCF is known to separate active from repressive chromatin marks[7,46,47]. We therefore investigated the possibility that suppressive H3K27me3 marks in the *Wap* locus of non-mammary cells would spread into the active *Ramp3* locus upon deletion of the CTCF sites. Based on ChIP-seq analyses[48] there were few, if any, H3K27me3 marks in the *Wap-Ramp3* locus of mammary tissue (Fig. 9a). The *Hoxα*

locus served as a positive control (Fig. 9b). Notably, in cerebella the *Wap* locus was also devoid of H3K27me3 marks (Fig. 9c), suggesting the inactivity of the *Wap* gene cannot be contributed to the presence of known suppressive marks. The *Hoxα* locus served as a positive control (Fig. 9d). Thus, silencing of *Ramp3* in non-mammary tissues from mutant mice lacking the CTCF sites cannot be explained by the spreading of known silencing chromatin marks[47,49–52].

**Discussion**

Although a fundamental role of CTCF in shielding genetic neighbourhoods in ESCs had been established[7,18], its ability to safeguard exceptionally strong cell-specific super-enhancers had not been investigated. Nor was it known whether CTCF enabled cell-specific genes to maintain a state of silence in non-target cells. Our findings emphasize more complex, and likely cell-specific, roles of CTCF sites and insulated neighbourhoods at the mammary-specific *Wap* locus that is activated by a cytokine-sensing super-enhancer during pregnancy[6]. Our *in vivo* study demonstrates, for the first time, that a cell-specific super-enhancer extends its functional reach beyond its respective target and past several CTCF sites into a juxtaposed neighbourhood where it activates a secondary target gene. Although these CTCF sites, including the loop anchor, do not block the super-enhancer, their deletion from the mouse genome demonstrates the capacity to muffle gene activation. We also demonstrate that these CTCF sites are essential for the expression

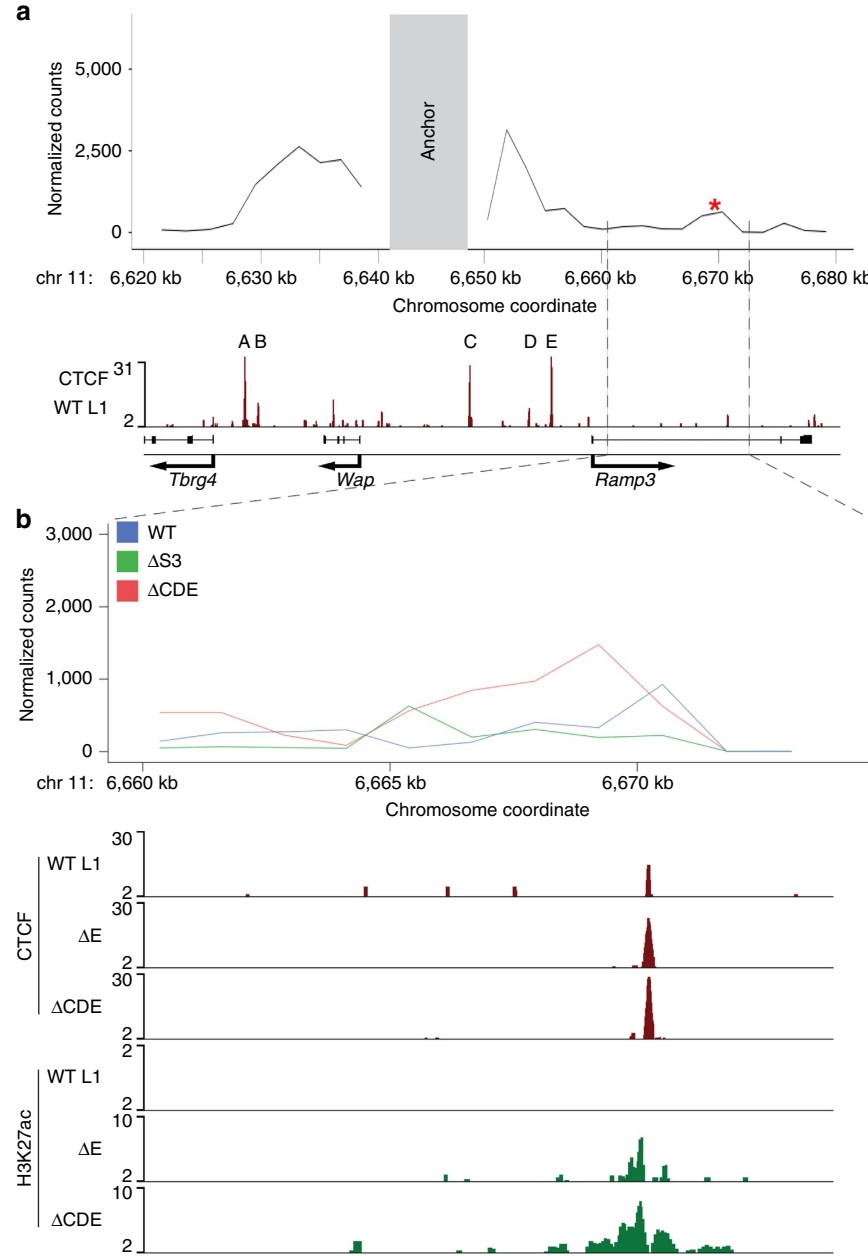

**Figure 6 | Interactions between the *Wap* super-enhancer and the first intron of *Ramp3*.** (a) 4C data at the *Tbrg4-Wap-Ramp3* locus were obtained from wild type lactating mammary tissue. The anchor region located over the third enhancer (S3) is shaded in grey. The lower panel shows CTCF-binding sites in the *Tbrg4-Wap-Ramp3* locus. 4C interactions occurred between the first intron of *Ramp3* and the anchor region (red asterisk). (b) 4C signal within the first intron of *Ramp3*. Intronic 4C interactions in wild type (blue) were absent in ΔS3 mutants (green) and increased in ΔCDE mutants (red). The lower panel shows increased CTCF binding to the intronic site in ΔE and ΔCDE mutants along with an increase in H3K27ac.

of the neighbouring *Ramp3* gene in non-mammary cells but are not required for the maintenance of transcriptional silence of *Wap* in non-mammary cells.

Although deletion of CTCF sites separating enhancers from non-target genes can result in their activation, it appears that in ESCs[7] mainly those genes respond that are expressed at very low levels (FPKM/RPKM)[7,10,16,18,53]. In contrast, the mammary super-enhancer not only activates its native *Wap* gene but also bypasses several CTCF sites to further activate *Ramp3*, a secondary target gene. This secondary target gene is located in a separate insulated neighbourhood and expressed across cell types. Thus, CTCF sites can muffle, but do not block, very strong enhancers. Moreover, loss of CTCF sites permits mammary enhancers to only induce genes that already display some baseline

activity but not silent genes[30]. Although the activation of genes upon loss of CTCF sites is generally attributed to *de novo* promoter–enhancers interactions[16,28,30,54–56], the mammary enhancer in this study establishes interactions with a dormant CTCF-binding site within the first intron of the neighbouring genes, which leads to the deposition of activating H3K27ac marks. While it is not clear why CTCF sites and loop anchors vary in their insulating strength[57], the presence of several CTCF sites might increase the "insulation score"[8]. Two hundred and seventy out of the 440 mammary-specific super-enhancers have more than one CTCF site as a boundary to the neighbouring genes, which would increase their insulation score.

The ability of enhancers to activate genes over great distances and independent of their location with respect to promoters is a

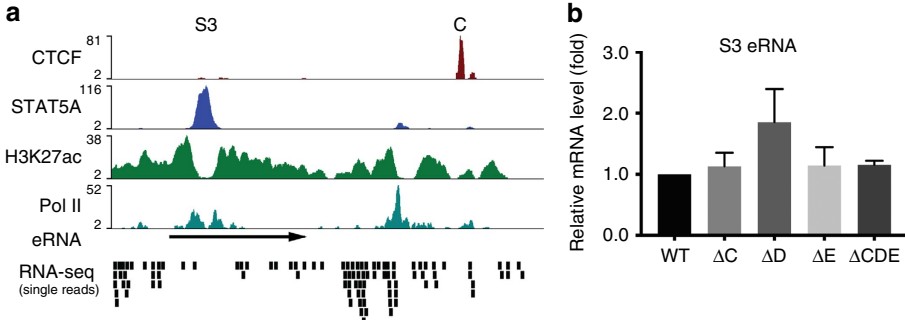

**Figure 7 | *Wap* super-enhancer eRNA levels are independent of CTCF sites. (a)** Structure of the S3 region of the super-enhancer. Site S3 is characterized by STAT5 binding, extensive H3K27ac, Pol II binding (GSE74826) and the presence of eRNAs (based on total RNA-seq data). Transcripts coincided with site S3 and additional STAT5-binding sites. An eRNA associated with S3 was identified and characterized using 5′ and 3′ RACE (Supplementary Note 2). **(b)** No statistically significant differences in S3 eRNA levels between mammary tissue from wild type and the mutants were observed. One-way ANOVA with multiple comparison and alpha 0.01 was applied ($P = 0.26$). Results are shown as the means ± s.e.m. of independent biological replicates; ΔC $n = 9$ ($-/-$); ΔD $n = 5$ ($-/-$); ΔE $n = 6$ ($-/-$); ΔCDE $n = 3$ ($-/-$).

core dogma[58] that is largely based on studies conducted outside the native genomic environment. However, *in vivo* rules guiding enhancers to pair up with their respective target genes are more complex[58] and the 'search space' of enhancers might be confined to TADs and insulated neighbourhoods[59,60]. Notably, while the *Wap* super-enhancer activates its own gene and the juxtaposed *Ramp3* located in a separate neighbourhood, it has little or no capacity to reach over its own target and activate the downstream *Tbrg4* gene even upon deletion of the two separating CTCF sites. This suggests that the search space of enhancers might not extend past its nearest promoters[61–63].

Boundaries that establish insulated neighbourhoods are required to maintain repression of specific gene sets in ESCs[7] and deletion of anchor CTCF sites leads to their inappropriate activation[7]. Although this mechanism is operative in Polycomb-repressed genes in ESCs, it does not apply to the mammary-specific *Wap* gene and possibly not to other genes controlled by mammary super-enhancers. While genes regulated by mammary-specific enhancers are silent in non-mammary cells, they do not display visible repressive H3K27me3 marks in non-expressing cells[48]. Loss of the CTCF neighbourhood anchor at the widely expressed *Ramp3* gene fails to awaken the juxtaposed silent *Wap* gene in non-mammary cell types, but it rather leads to the suppression of *Ramp3* expression. Thus, CTCF sites bordering the mammary super-enhancer serve two distinct functions, limiting its exposure to the neighbouring *Ramp3* gene in mammary epithelium and permitting *Ramp3* expression in non-mammary cells. At this point it is not clear whether CTCF boundaries associated with other cell-specific super-enhancers also exhibit such dual functions. In summary, our study suggests more complex and cell-specific functions of CTCF sites associated with exceptionally strong lineage-restricted super-enhancers.

## Methods

**Generation of mutant mice.** CRISPR/Cas9 targeted mice were generated by the transgenic core of the National Heart, Lung, and Blood Institute (NHLBI). Single-guide RNAs (sgRNA) were designed using MIT's CRISPR Design tool (http://crispr.mit.edu/). Those spanning CTCF-binding sites were synthesized (OriGene, Thermo Fisher Scientific). The sgRNAs were injected with Cas9 protein into the cytoplasm of fertilized eggs from C57BL/6 × C57BL/6 mating and transplanted into pseudopregnant C57BL/6 mice to obtain founders. Sequences of the sgRNA used can be found in Supplementary Table 1. Mice carrying *loxP* sites flanking CTCF sites A and B were generated by InGenious using homologous recombination. The sequences between loxP sites were deleted using the germline Figla-Cre delete mice (gift from Jurrien Dean, NIDDK, NIH).

**Genotyping and generation of homozygous mutant mice.** All animal procedures were in accordance with NIH, NIDDK guidelines for the care and use of laboratory animals. Founders, many of them mosaic, were bred with C57BL/6 wild type mice to segregate the various mutant alleles. Tail snips were taken from mice and genotyped by genomic DNA PCR. The PCR products were analysed by Sanger sequencing. Primer information can be found in Supplementary Table 2. Once mutant alleles were identified and characterized, F1 mice were inbred to obtain homozygous mice. The mouse lines generated by CRISPR/CAS9 gene editing and used in this study carried the following deletions: line ΔC (58 bp), ΔD (134 bp), ΔE (160 bp) and ΔCDE (ΔC 2,442 bp; ΔD 4 bp; ΔE 1,165 bp). The deletion of line ΔAB is 1,711 bp. The specific deletion of each line can be found in the Supplementary Note 1.

**RNA isolation and qRT-PCR.** Total RNA was isolated using the PureLink RNA Mini kit (Ambion) per the manufacturer's protocol and cDNA was synthesized from 1 μg RNA using oligo dT primers and SuperScript II/III (Invitrogen). Quantitative TaqMan PCR was performed in triplets on the CFX384 Real-Time PCR Detection System (Bio-Rad) using the following probes: *Tbrg4* Mm01220234_g1; *Wap* Mm00839913_m1; *Ramp3* Mm00840142_m1; *Gapdh* 4352339E of Applied Biosystems.

Ct-values were normalized to the housekeeping gene *Gapdh* and fold-changes were calculated using the comparative Ct-method ($2^{-\Delta\Delta Ct}$ method).

**RACE.** RACE assay was done using the FirstChoice RLM-RACE Kit (Ambion) according to the manufacturer's instructions. The following primers were used in the nested PCRs in the assay.

5′RACE, Outer Primer 5′-CTCTTCCACCCTGTCCACTGCTC-3′; Inner Primer 5′-AGAGTTGATGGGGCAGGAAAGAGCC-3′.

3′RACE, Outer Primer 5′-CACATAGTAGCCGAGGATGGCC-3′; Inner Primer 5′-GGCACCTGCCTCCTCCTTCTAGTCT-3′.

The PCR products were then purified and sequenced to identify the 5′ and 3′ end of eRNA transcript (Supplementary Note 2).

**ChIP-seq.** Mammary tissues were collected on day one of lactation (L1) and cerebella from 2- to 11-month-old female mice. For cerebellum ChIP, four cerebella were pooled per biological replicate. All tissues were snap frozen and stored at −80 °C. Tissues were pulverized in liquid nitrogen followed by chromatin fixation with 1% formaldehyde final concentration. Washed chromatin (PBS) was homogenized using a dounce tissue grinder in Farnham's lysis buffer and followed by chromatin fragmentation through sonication. Fragmentation efficiency in the range of 200–500 bp was controlled by gel electrophoresis and ChIP was performed using 50 μl Dynabeads Protein A (Invitrogen) and 10 μg of the following antibodies per ChIP (Manufacturer, Catalogue Number): anti-CTCF (Abcam, ab70303), anti-H3K27ac (Abcam, ab4729), anti-H3K27me3 (Abcam, ab6002). Ten microlitres of anti-CTCF (Millipore, 07-729) was used (no concentration provided by Millipore). Indexed libraries were produced using the NEBNext Ultra II DNA Library Prep Kit and NEBNext Index Primer Set for Illumina according to the manufacturer's protocol. Fragments between 200 and 500 bp were selected by gel electrophoresis and 50 bp single-end sequencing was performed on an Illumina HiSeq 2500.

**Total RNA-seq.** Total RNAs were extracted from mammary tissue at day 18 of pregnancy and purified twice with Trizol and RNeasy Plus Mini Kit (Qiagen, 74134). Ribosomal RNA was removed from 1 μg of total RNAs and cDNA was

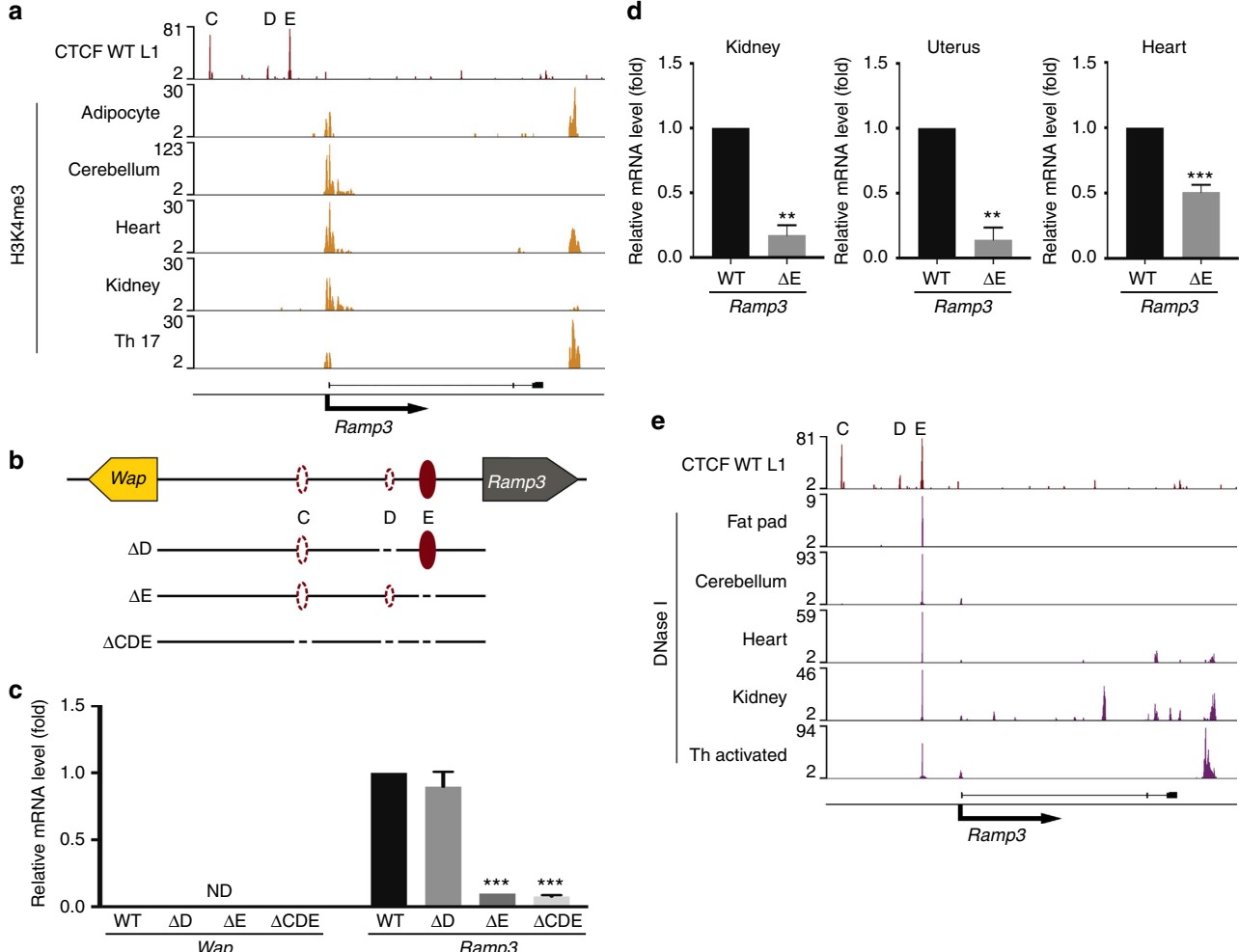

**Figure 8 | Border CTCF site is required for *Ramp3* expression in non-mammary cells. (a)** H3K4me3 marks at the *Ramp3* promoter indicated active transcription in several non-mammary cell types and tissues (GSE29218, GSE31039, GSE32864). (**b**) Out of the three CTCF-binding sites in mammary tissue that separate the *Wap* super-enhancer and *Ramp3*, only site E (filled crimson ovals) was present in non-mammary tissue, like cerebellum, heart, kidney and uterus. CTCF binding to sites C and D was absent in those tissues (dashed crimson ovals) (**c**) Deletion of site E as well as the combined deletion of sites C, D and E led to a 90% reduction in *Ramp3* expression in the cerebellum. Deletion of site D did not affect *Ramp3* expression. Results are shown as the means ± s.e.m. of independent biological replicates; ΔD: $n = 3$ ( − / − ); ΔE: $n = 6$ ( − / − ); ΔCDE: $n = 4$ ( − / − ); data were normalized to *Gapdh*; one-way ANOVA with multiple comparison and alpha 0.01 was applied. *$P < 0.01$, **$P < 0.001$, ***$P < 0.0001$. (**d**) Deletion of site E led to a significant reduction in *Ramp3* expression in several non-mammary cells. Results are shown as the means ± s.e.m. of independent biological replicates; Kidney and Uterus ΔE: $n = 3$ ( − / − ); Heart ΔE: $n = 4$ ( − / − ); data were normalized to *Gapdh*; a two-tailed Student's *t*-test with alpha 0.01 was applied. *$P < 0.01$, **$P < 0.001$, ***$P < 0.0001$. (**e**) CTCF site E coincided with DNase I hypersensitivity. Sites C and D were not occupied by CTCF in non-mammary cells and no DNase I hypersensitivity had been detected at these sites (GSE37074).

synthesized using SuperScript II (Invitrogen). Libraries for sequencing were prepared according to the manufacturer's instructions with TruSeq Stranded Total RNA Library Prep Kit with Ribo-Zero (Illumina, RS-122-2201) and paired-end sequencing was done with a HiSeq 2000 instrument (Illumina).

**Circular chromosome conformation capture (4C)-seq.** The 4C protocol was adapted from published methods[64,65]. Frozen-stored mammary tissues collected at L1 were ground into powder. Chromatin was fixed with formaldehyde (final 1%) for 15 min at room temperature, and then was quenched with glycine (final 0.125 M). Pellets were lysed in lysis buffer (10 mM Tris HCl pH 8.0, 10 mM NaCl, 0.5% Nonidet P-40) containing phenylmethylsulfonyl fluoride and protease inhibitors, incubated on ice for 30 min, and dounced using a pre-chilled glass homogenizer, followed by another 10 min incubation on ice. After removal of supernatant, nuclei pellets were re-suspended in DpnII buffer (New England Biolabs). SDS was added to a final concentration of 0.2% and samples were incubated for 1 h at 37 °C. Two per cent Triton X-100 was added, followed by incubation for 1 h at 37 °C. Samples were incubated with 400 units each of restriction enzymes DpnII (New England Biolabs) and were incubated overnight at 37 °C. To enzyme inactivation SDS was added (final 1%) and incubated for 30 min at 65 °C, followed by SDS-sequestration with 1% Triton X-100 at 37 °C for 1 h, T4

ligase (New England Biolabs) was added to each sample and incubated for overnight at 16 °C. Ligated samples were treated overnight with proteinase K (20 mg ml − 1, Invitrogen) at 65 °C and 1 h at 37 °C with RNase A (10 mg ml − 1, Thermo Fisher Scientific), and DNA fragments were purified by phenol–chloroform method. DNA fragments from 3C were digested with *Mse*I (New England Biolabs) overnight at 37 °C and ligated using T4 ligase overnight at 16 °C. DNA was purified by phenol/chloroform method and then amplified with site-specific primers linked to the Illumina DNA adaptors. Libraries for next generation sequencing were prepared and sequenced with HiSeq 2500 (Illumina).

**ChIP-seq analysis.** ChIP-seq data were analysed using Trimmomatic[66] (version 0.33), for quality check and removal of low-quality reads (parameters: LEADING: 20, TRAILING: 20, SLIDINGWINDOW: 4:20, MINLEN: 20, HEADCROP: 15), and Bowtie aligner[67] (version 1.1.2) with the − m 1 parameter to obtain only uniquely mapped reads, except for CTCF samples where the − m 3 and best parameters were used. Reads were mapped to the reference genome mm10.

For ChIP-seq data from GEO, Trimmomatic[66] (version 0.33) was used to check read quality (using the following parameters: LEADING: 3, TRAILING: 3, SLIDINGWINDOW: 4:20, MINLEN: 20). The alignment was performed applying Bowtie aligner[67] (version 1.1.2) using − m 1 as the parameter.

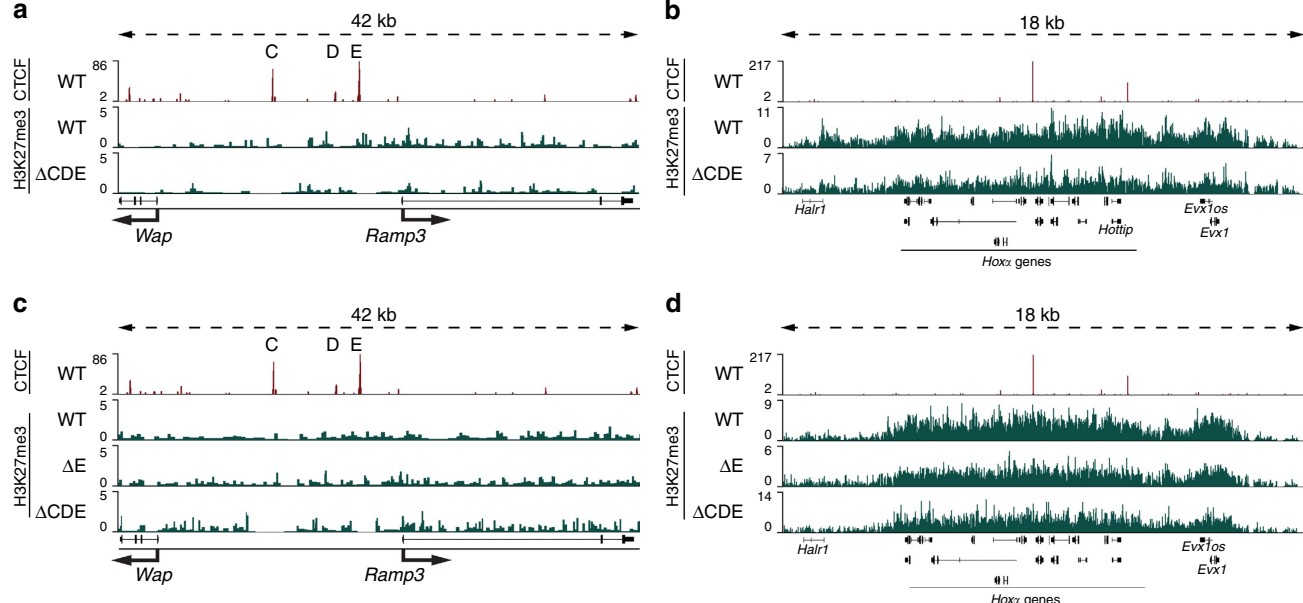

**Figure 9 | Absence of K3K27me3 marks in the *Wap-Ramp3* locus in mammary tissue and cerebellum. (a)** H3K27me3 marks were virtually absent in the *Wap-Ramp3* locus in mammary tissue from WT and ΔCDE mutants. **(b)** The *Hoxα* locus served as a control locus and showed extensive H3K27me3 coverage in mammary tissue. **(c)** No increase of H3K27me3 marks in ΔE and ΔCDE mutants in the cerebellum. **(d)** The *Hoxα* locus served as a control locus in the cerebellum and showed strong H3K27me3 coverage.

HOMER software[68] (default settings) and Integrative Genomics Viewer[69] (IGV) were applied for visualization. MACS2 (ref. 70) peak finding algorithm (version 2.1.0) was used to identify regions of ChIP-seq enrichment over background. For further analyses additional tools were used[68,71,72].

**DNase-seq analysis.** DNase-seq data were treated as described[73]. The reads were trimmed to the first 20 bp using Trimmomatic[66] and bwa[74] (default setting) was used for alignment to mm10. Further visualization was done with Homer[68] and IGV[69].

**mRNA-seq analysis.** RNA-seq data were trimmed using Trimmomatic[66] (parameters: LEADING: 20, TRAILING: 20, SLIDINGWINDOW: 4:20, MINLEN: 20, HEADCROP: 15) and mapped using a STAR RNA-seq aligner[75] (default settings, GRCm38.84 as a GTF file). The GTF file was filtered, by excluding predicted genes (LOC, Rik and BC), to retain only high-confident genes. RNA-seq analyses were done using R (version 3.2.3), Bioconductor[76], and the packages Rsubread[77] and DESeq2 (ref. 78).

**Total RNA-seq analysis.** Total RNA-seq reads were analysed using Trimmomatic[66] (version 0.33) to check read quality (with following parameters: LEADING: 3, TRAILING: 3, SLIDINGWINDOW: 4:20, MINLEN: 36). The alignment was performed in the Bowtie aligner[67] (version 1.1.2) using paired end mode.

**4C-seq data analysis.** The sequencing data were processed using the 4C-ker[79]. All 4C-seq images were generated using 4Cker R package with $k = 4$ and $k = 6$.

**Statistical analyses.** The samples for qRT–PCR and ChIP-seq were randomly, but not blinded, selected. Statistical analysis of qRT–PCR and RNA-seq data was performed as follows. Data were analysed for normal distribution using Shapiro–Wilk normality test and statistical power was calculated with a significance level of 0.01 and a statistical power of 0.9. Power calculations were adapted for two-sample *t*-tests and one-way ANOVA analyses. Statistical significance between two groups was calculated using a two-tailed Student's *t*-test with a confidence level of 99%. For comparison between more than two groups, one-way ANOVA was used with a confidence level of 99%. Analyses were done in R and GraphPad Prism (version 7.0a).

**URLs.** MIT CRISPR Design tool, http://crispr.mit.edu/; R Project for Statistical Computing, https://www.R-project.org/ and dplyr (https://CRAN.R-project.org/package=dplyr).

**Data availability.** ChIP-seq, RNA-seq and 4C-seq data are available at NCBI's Gene Expression Omnibus (GEO). The SuperSeries is accessible under GSE92932, comprising ChIP-seq under the accession number GSE92587, RNA-seq under GSE92931 and 4C-seq under GSE97803.

Already published ChIP-seq, DNase-seq and RNA-seq data were taken from GEO: GSE29218, GSE31039, GSE32864, GSE37074, GSE37646, GSE40918, GSE74189, GSE74826, GSE92590, GSE90788. Hi-C and ChIA-PET data are available under GSE35156 and GSE57911, respectively.

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

## Acknowledgements

We thank Teresa Mohr for genotyping mice, Christian Reinbold for performing eRNA and qRT-PCR assays, as well as Chithra Keembiyehetty Nightingale, Sijung Yun and Harold Smith from the NIDDK Genomics Core for NGS and Chengyu Liu from the

NHLBI mouse core for generating mutant mice. We thank Ann Dean, Gary Felsenfeld and Raphael Casellas for discussions. M.W. is a graduate student of the Individual Graduate Partnership Program (GPP) between NIH/NIDDK and the Medical University of Innsbruck. This work was performed in partial fulfilment of the graduation requirements for M.W and the master thesis of F.R. We thank Z. Trajanoski for advising M.W. during her graduate studies.

This work was supported by the IRP of the NIDDK, NIH and by a grant of the Korean Health Technology R&D Project, Ministry of Health & Welfare, Republic of Korea (HI15C1184) to H.K.L.

## Author contributions

M.W. designed experiments and supervised the project, analysed ChIP-seq, RNA-seq and 4C data, performed computational and statistical analyses, and wrote the manuscript. K.H.Y. initiated and supervised the project, identified mutant founder mice and established mouse lines, conducted ChIP-seq and expression studies. F.R. conducted mouse work, expression and ChIP-seq studies. T.M.K. identified founder mice, established lines and conducted ChIP-seq and expression studies. C.W. conducted gene expression and eRNA expression as well as ChIP-seq experiments, provided expertise and feedback. H.K.L. conducted gene expression and 4C experiments. L.H. conceived, designed and supervised the study, analysed data and wrote the manuscript. M.W. and L.H. wrote the manuscript, which was approved by all authors.

## Additional information

**Competing interests:** The authors declare no competing financial interests.

