## [Peer Review File · Nature Communications]

Reviewers' comments:

Reviewer #1 (Remarks to the Author):

The question addressed in this study is whether "the search space of cell-specific super-enhancers is constrained by CTCF" using the Wap gene, and the neighbouring genes Tbrg4 and Ramp3, as a model. Wap is highly expressed in the lactating mouse mammary gland and controlled by a mammary specific superenhancer. Building on their previous article in Nature Genetics, they now show that expression of Ramp3 (like Wap) increases in the lactating mouse mammary gland (compared to day 6 of pregnancy) via a mechanism that is dependent on the E3 enhancer within the Wap superenhancer and that deletion of the CTCF binding site closest to the 5' end of Ramp3 induces further expression of Ramp3, without altering the levels of H3K27ac or eRNA within the Wap superenhancer, nor spreading of H3K27ac towards the Ramp3 gene. However, deletion of the same CTCF binding sites have different effects on Ramp3 and Wap expression in the cerebellum, illustrating how tissue-specific differences in gene expression may be achieved.

Overall these results are interesting and well supported, but also appear of a rather restricted and preliminary nature as delineated in more detail below.

1. The results are limited to an analysis of effects of a single mammary specific superenhancer on a single gene. This makes it difficult to infer the extent to which this finding may reflect a more generalized mechanism of cell-type specific control of gene expression.
2. It would be important to also establish whether the deletion effects are direct or indirect. Is the Ramp3 gene in direct contact with the "E3" enhancer? And is this contact disrupted when the E3 enhancer is deleted and/or strengthened by the deletion of the CTCF site? The authors could perform 3C type experiments in these samples, to show a direct contact between these 2 genomic regions, and strengthen the inference that the effects seen are direct.
3. Does deletion of the E3 enhancer produce differences in chromatin marks at the Ramp3 promoter?
4. Does the removal of the CTCF binding site correlate with a reduction in "inhibitory marks" around the CTCF site and/or the Ramp3 promoter? An alternative explanation to their results could be that the removal of the CTCF site removes a transcriptional block that is independent of the E3 enhancer. On the same note, does removal of the E3 enhancer counteract the increased Ramp3 expression caused by the removal of the CTCF binding site?
5. The samples used for the analysis were extracts of whole tissue, containing many different cell types. Therefore the authors should at least acknowledge this as a possible confounding factor in interpreting their findings. For example, given that Ramp3 is only modestly upregulated during lactation compared to Wap and the Ramp3 promoter has low levels of all activating marks measured (Fig 1a), upregulation of Wap and Ramp3 during lactation could occur in distinct cell types, where the E3 enhancer is alternatively used to regulate each gene individually.

Reviewer #2 (Remarks to the Author):

The manuscript addresses a very interesting issue in CTCF biology i.e. the ability of CTCF to modulate enhancer-promoter interactions in an in vivo setting. The manuscript is in principle appropriate for Nat Comm but the authors should first consider the following points:

1. On page 2, authors state that "CTCF.....has been shown to be instrumental in establishing such insulated neighborhoods and is also enriched at TADs, both known to confine enhancer activity". CTCF is enriched at TAD borders, not really TADs. Authors should consider that the insulated neighborhoods are CTCF loops that are part of TADs. The two are not necessarily different.
2. Page 2 "we turned to the mammary genome". This is an awkward statement at many different

levels. The genome of cells in the mammary gland is the same as in most other cells in the organism. Authors should mention that they are performing this analysis in mice.

3. Page 2. In the paragraph starting "In a quest to understand the physiological role of CTCF", authors should make clear what cells the different statements apply to. What cells were used to perform the ChIP with CTCF? Are these the same cells where Wap is expressed by the super-enhancer? It would be nice if the authors include the orientation of the CTCF sites shown in Figures 1A and/or 2A.

4. Page 3. "Since biologically significant CTCF sites are likely conserved across cell types". This statement has nothing to do with the publications cited and it is likely incorrect. CTCF sites conserved across cell types are probably involved in the expression of genes common to those cell types such as housekeeping genes.

5. Page 3. "CTCF binding to sites A, which coincides with a TAD boundary. The resolution in the Hi-C data from Dixon et al was around 50 kb, making it impossible to map the TAD boundary specifically to the A site. Also, where it says "preferentially found in cell types" the authors may mean "preferentially found in specific cell types"?

6. Page 3. "To clarify to what extent any of these CTCF sites serve as roadblocks and obstruct Wap super-enhancer activity". It is not really understood how CTCF interferes with enhancer-promoter interactions. The term "roadblock" has a very specific mechanistic implication that I'm not sure is supported by any experimental results.

7. Page 3. "Deletion of these sites in the mouse germline resulted in the loss of CTCF binding (Figure 2c) but did not result in an altered expression of Tbrg4 or Wap in lactating mammary tissue". Was the loss of CTCF binding also determined in lactating mammary tissue?

8. Page 3. "This is in accordance with studies in other systems^{20,21} and suggests that the Wap super-enhancer is unable to extend its sphere of influence past the Wap gene." There appear to be several CTCF sites within the Wap gene that, surprisingly, the authors seem to ignore. One of these sites becomes very prominent in the DeltaAB mutant, and seems to flank the super-enhancer. Could this be the reason for why the super-enhancer cannot activate the Tbrg4 gene in this mutant, because the super-enhancer is still flanked by CTCF in spite of the deletion of sites A and B?

9. Figure 3C. The DeltaC sample has more H3K27ac signal between the C and E CTCF sites. However, it seems that the signal to the left of C is also higher. How was the signal for H3K27ac normalized among the different samples in this figure? Since WT and DeltaD also have signal in this region, it could be that the extra signal in DeltaC is a normalization issue.

10. Page 4. "demonstrating that the inability of Ramp3 regulatory elements to spread into the Wap locus". Should it read "demonstrating the inability of Ramp3 regulatory elements to spread into the Wap locus" instead?

11. Page 4. Authors should comment on why deletion of CTCF site E in the cerebellum cause a decrease in Ramp3 expression. The argument that there are regulatory elements in the region is not convincing. These elements are very close to the promoter and presumably they can interact with the promoter without CTCF. Is it possible that there is another enhancer to the right of Ramp3 but outside the region shown in Figure 3? Knowing the orientation of site E would help understand where it may be looping, to the left or to the right.

12. Page 4. "Experimental evidence that TAD boundaries and chromatin loops anchored by CTCF". If a TAD boundary has a CTCF site, it because this site is an anchor for a loop. So the two are the same.

13. Page 5. "The mammary-specific Wap super-enhancer activates the two juxtaposed promoters, but has little or no capacity to effectively reach over its native target gene to activate the downstream gene, even upon deletion of separating CTCF sites". Please see concern expressed in #8 above about the possible role of additional CTCF sites within the Wap gene. I'm not sure this conclusion is supported by the data given the high occupancy of one of these CTCF sites in the DeltaAB mutant.

14. Page 5. "Our finding that an anchor CTCF site serves a dual purpose, muffling a superenhancer and coincides with cell-specific regulatory features, suggests a more complex biological role for these elements". It's not clear what the authors mean by this. I don't think that any of the results in the manuscript imply a more complex role for CTCF sites.

Reviewer #1

The question addressed in this study is whether "the search space of cell-specific super-enhancers is constrained by CTCF" using the Wap gene, and the neighbouring genes Tbrg4 and Ramp3, as a model. Wap is highly expressed in the lactating mouse mammary gland and controlled by a mammary specific superenhancer. Building on their previous article in Nature Genetics, they now show that expression of Ramp3 (like Wap) increases in the lactating mouse mammary gland (compared to day 6 of pregnancy) via a mechanism that is dependent on the E3 enhancer within the Wap superenhancer and that deletion of the CTCF binding site closest to the 5' end of Ramp3 induces further expression of Ramp3, without altering the levels of H3K27ac or eRNA within the Wap superenhancer, nor spreading of H3K27ac towards the Ramp3 gene. However, deletion of the same CTCF binding sites have different effects on Ramp3 and Wap expression in the cerebellum, illustrating how tissue-specific differences in gene expression may be achieved.

Overall these results are interesting and well supported, but also appear of a rather restricted and preliminary nature as delineated in more detail below.

1. The results are limited to an analysis of effects of a single mammary specific superenhancer on a single gene. This makes it difficult to infer the extent to which this finding may reflect a more generalized mechanism of cell-type specific control of gene expression.

Response

Yes, since the findings are from one super-enhancer and the CTCF sites associated with it, it is not possible to establish a generalized mechanism.

It would take several years to conduct an additional and equally comprehensive study and this would be beyond the scope of this manuscript. The current study included the generation of five different mouse lines carrying various deletions of different CTCF sites, individually and in relevant combinations. As we have outlined in a manuscript that has been accepted at *Nature Communications*, deleting juxtaposed sites required sequential targeting.

2. It would be important to also establish whether the deletion effects are direct or indirect. Is the Ramp3 gene in direct contact with the "E3" enhancer? And is this contact disrupted when the E3 enhancer is deleted and/or strengthened by the deletion of the CTCF site? The authors could perform 3C type experiments in these samples, to show a direct contact between these 2 genomic regions, and strengthen the inference that the effects seen are direct.

Response

We have now addressed this question and conducted 4C (3C-seq) experiments. We have used the S3 region of the super-enhancer as an anchor and conducted experiments in WT tissue and in tissues lacking either the S3 enhancer or the three CTCF sites separating the *Wap* super-enhancer from the *Ramp3* gene. We have detected interactions between the S3 enhancer region and the first intron of the *Ramp3* gene (new Figure 7). This area also acquired H3K27ac marks (new Figure 5a and Figure 7) and we propose that this triggers increased expression of *Ramp3*. Of note: to avoid confusions, we changed the enhancer name from E3 to S3 (binding of STAT5). This was also the original name in the Nature Genetics paper. E refers to the CTCF site that anchors the *Ramp3* loop.

3. Does deletion of the E3 enhancer produce differences in chromatin marks at the *Ramp3* promoter?

Response

In WT mammary tissue only very little, if any, H3K27ac is detected at the *Ramp3* promoter (Figure 3d). Similarly the degree of H3K27ac in enhancer S3 mutant tissue is below the threshold of detection (Figure 3d). There was also no detectable H3K27ac at the *Ramp3* promoter in other cell types (Th17 and adipocytes) in which *Ramp3* is expressed (Figure 3e). We have also included positive controls of genes preferentially expressed in these cells (Figure 3f and g).

4. Does the removal of the CTCF binding site correlate with a reduction in "inhibitory marks" around the CTCF site and/or the *Ramp3* promoter? An alternative explanation to their results could be that the removal of the CTCF site removes a transcriptional block that is independent of the E3 chromatin marks enhancer. On the same note, does removal of the E3 enhancer counteract the increased *Ramp3* expression caused by the removal of the CTCF binding site?

Response

We addressed this question by analyzing H3K27me3 patterns in wild type mammary tissue and in mutant tissue lacking all CTCF sites separating the *Wap* super-enhancer from *Ramp3*. There is little H3K27me3 in the *Wap-Ramp3* locus and the pattern does not change in mutant tissue (Figure 9). The *Hox* locus served as a positive control for the ChIP-seq experiment.

Yes, it would be possible that deletion of the CTCF sites would release a block that is independent of the S3 enhancer. To further address this issue we have now conducted 4C (3C-seq) experiments and identified enhanced interactions between the S3 enhancer and the *Ramp3* gene in mutant tissue (Figure 7). We have also observed increased H3K27ac in the first intron *Ramp3*, which coincides with enhanced CTCF binding to a pre-existing site (Figure 5 and

Figure 7).

The reviewer asked whether “removal of the S3 enhancer counteract the increased *Ramp3* expression caused by the removal of the CTCF binding site?” At this point we do not have mice lacking both the S3 enhancer site and the CTCF site. These mice cannot be generated by breeding the two individual knock-outs with each other and we would need to introduce the S3 enhancer mutation in the background of the CTCF mutations (or the other way around). Although feasible such an experiment would take approximately one year and we believe that it is outside the scope of this study. Primary cells, or cell lines, cannot be used for such experiments since mammary enhancers are only established and active in intact tissue during pregnancy within the living organism.

5. The samples used for the analysis were extracts of whole tissue, containing many different cell types. Therefore the authors should at least acknowledge this as a possible confounding factor in interpreting their findings. For example, given that *Ramp3* is only modestly upregulated during lactation compared to *Wap* and the *Ramp3* promoter has low levels of all activating marks measured (Fig 1a), upregulation of *Wap* and *Ramp3* during lactation could occur in distinct cell types, where the E3 enhancer is alternatively used to regulate each gene individually.

Response

The vast majority of cells in mammary tissue during lactation are milk secreting alveolar cells and they display active mammary enhancers. Other known cell types in mammary tissue, including adipocytes (Kang 2013) and immune cells (Kang 2013) do not feature activating marks on mammary enhancers. We have now included data that demonstrate that neither *Wap* nor *Ramp3* display H3K27ac marks on the S3 region in Th17 cells and adipose cells (Figure 3e). Since there is no evidence that the S3 region carries active histone marks in any non-mammary cell type it is likely that *Ramp3* is active in secreting alveolar epithelial cells. To demonstrate to what extent *Ramp3* is expressed in non-alveolar cells in mammary tissue, and possibly regulated by CTCF sites, we analyzed tissue from non-parous (virgin) mice (Figure 4e). While there is little or no *Wap* mRNA (Ct >35), *Ramp3* is expressed at low levels (Ct ~33). Moreover, the combined loss of CTCF sites C, D and E did not affect *Wap* expression in non-parous mice but *Ramp3* RNA levels decreased by ~75%. This is equivalent to the reduction of *Ramp3* upon loss of these CTCF sites in other tissues and suggests that they contribute to *Ramp3* regulation in several cell types. All this has now been discussed. The vastly higher levels of *Wap* mRNA compared to *Ramp3* could be the result of additional regulatory elements driving *Wap* expression or differential mRNA stabilities. This has now been discussed.

Reviewer #2:

The manuscript addresses a very interesting issue in CTCF biology i.e. the ability of CTCF to modulate enhancer-promoter interactions in an in vivo setting. The manuscript is in principle appropriate for Nat Comm but the authors should first consider the following points:

1. On page 2, authors state that “CTCF.....has been shown to be instrumental in establishing such insulated neighborhoods and is also enriched at TADs, both known to confine enhancer activity”. CTCF is enriched at TAD borders, not really TADs. Authors should consider that the insulated neighborhoods are CTCF loops that are part of TADs. The two are not necessarily different.

Response

We corrected this and stated specifically that CTCF loops are part of TADs.

2. Page 2 “we turned to the mammary genome”. This is an awkward statement at many different levels. The genome of cells in the mammary gland is the same as in most other cells in the organism. Authors should mention that they are performing this analysis in mice.

Response

Yes, if one considers the genome purely as the DNA content then the term “mammary genome” is incorrect. We stated early on in the manuscript that our study was conducted in mice.

3. Page 2. In the paragraph starting “In a quest to understand the physiological role of CTCF”, authors should make clear what cells the different statements apply to. What cells were used to perform the ChIP with CTCF? Are these the same cells where *Wap* is expressed by the super-enhancer? It would be nice if the authors include the orientation of the CTCF sites shown in Figures 1A and/or 2A.

Response

We have now described that we have studied mammary tissue from lactating mice, which consists of ~90% alveolar epithelium, the cells that are characterized by mammary super-enhancers and express *Wap*. Other cell types in mammary tissue, such as adipocytes and immune cells do not have active mammary enhancers. Several years ago we have analyzed STAT5 ChIP-seq data from different cell types (Kang 2013) and mammary regulatory elements are recognized by STAT5 only in mammary tissue. We have included the orientation of CTCF sites.

4. Page 3. “Since biologically significant CTCF sites are likely conserved across cell types”. This statement has nothing to do with the publications cited and it is likely incorrect. CTCF sites conserved across cell types are probably involved in the expression of genes common to those cell types such as housekeeping genes.

Response

To address whether sites bound by CTCF in many tissues are associated with common genes and, conversely, mammary-specific ones, we have now globally analyzed CTCF binding across cell types, at mammary-specific enhancers and super-enhancers (new Figure 1). In general, almost all CTCF sites associated with mammary loci are recognized across cell types. There are no clear mammary-specific binding sites. In some cases binding is enriched in mammary cells but also detected in ESCs and T cells, cell types responsive to cells highly dependent on cytokine-STAT signaling. We have discussed this.

5. Page 3. “CTCF binding to sites A, which coincides with a TAD boundary. The resolution in the Hi-C data from Dixon et al was around 50 kb, making it impossible to map the TAD boundary specifically to the A site. Also, where it says “preferentially found in cell types” the authors may mean “preferentially found in specific cell types”?

Response

We have addressed this.

6. Page 3. “To clarify to what extent any of these CTCF sites serve as roadblocks and obstruct Wap super-enhancer activity”. It is not really understood how CTCF interferes with enhancer-promoter interactions. The term “roadblock” has a very specific mechanistic implication that I’m not sure is supported by any experimental results.

Response

To address how loss of CTCF sites results in the activation of *Ramp3*, we conducted 4C (new Figure 7) and H3K27ac ChIP-seq on mammary tissue (Figure 5a) from WT and mutant mice lacking the three CTCF sites that separate the *Wap* super-enhancer from the *Ramp3* gene.

7. Page 3. “Deletion of these sites in the mouse germline resulted in the loss of CTCF binding (Figure 2c) but did not result in an altered expression of *Tbrg4* or *Wap* in lactating mammary tissue”. Was the loss of CTCF binding also determined in lactating mammary tissue?

Response

Yes, we confirmed loss of CTCF binding in mutant mammary tissue during lactation (new Figure 4b).

8. Page 3. “This is in accordance with studies in other systems 20,21 and suggests that the *Wap* super-enhancer is unable to extend its sphere of influence past the *Wap* gene.” There appear to be several CTCF sites within the *Wap* gene that, surprisingly, the authors seem to ignore. One of these sites becomes very prominent in the DeltaAB mutant, and seems to flank the super-enhancer. Could this be the reason for why the super-enhancer cannot activate the *Tbrg4* gene in this mutant, because the super-enhancer is still flanked by CTCF in spite of the deletion of sites A and B?

Response

Yes, there is one CTCF site within the third intron of the *Wap* gene, which could serve as a boundary. However, loss of CTCF sites A and B does not necessarily result in stronger CTCF binding to this site (note the scale). We have discussed these data.

9. Figure 3C. The DeltaC sample has more H3K27ac signal between the C and E CTCF sites. However, it seems that the signal to the left of C is also higher. How was the signal for H3K27ac normalized among the different samples in this figure? Since WT and DeltaD also have signal in this region, it could be that the extra signal in DeltaC is a normalization issue.

Response

Scales for the H3K27ac density (reads) are shown in Figure 5c. We have analyzed at least two biological replicates from each genotype. To determine whether H3K27ac signals increase past the deleted CTCF sites, we have also analyzed H3K27ac signals over the *Wap* super-enhancer as control. The same can be applied for the triple mutation (DeltaCEF) where we observed increased K27ac over the *Ramp3* promoter and a specific region in the first intron. The signal was normalized to 10 million reads (Heinz 2010).

10. Page 4. “demonstrating that the inability of *Ramp3* regulatory elements to spread into the *Wap* locus”. Should it read “demonstrating the inability of *Ramp3* regulatory elements to spread into the *Wap* locus” instead?

Response

We have corrected this.

11. Page 4. Authors should comment on why deletion of CTCF site E in the cerebellum cause a decrease in *Ramp3* expression. The argument that there are regulatory elements in the region is not convincing. These elements are very close to the promoter and presumably they can interact with the promoter without CTCF. Is it possible that there is another enhancer to the right of *Ramp3* but outside the region shown in Figure 3? Knowing the orientation of site E would help understand where it may be looping, to the left or to the right.

Response

We have now marked the orientation of all CTCF sites and based on current knowledge, site E could interact with site F, at a distance of just over 100 kb (new Figure 2a). Published Hi-C data from Lieberman-Aiden and ChIA-PET data demonstrate such a loop.

The reviewer is correct that we do not have additional experimental evidence that CTCF site E coincides with regulatory elements. Although, ChIP-seq data from cerebellum demonstrate the presence of H3K4me1 marks in this area this is probably not sufficient evidence.

Ramp3 expression declined in cerebella lacking CTCF site E and CDE combined. We have now analyzed *Ramp3* expression in other cell types. Measurable levels of *Ramp3* were detected in mammary tissue from virgin mice (composed mainly of adipocytes and other non-mammary cells), heart, kidney and uterus. *Ramp3* expression in these tissues also declined in the absence of CTCF site E. There are two possibilities to explain these findings. First, a positive regulatory element coincides with CTCF site E that activates the *Ramp3* promoter in non-mammary cells. Alternatively, loss of CTCF site E exposes the *Ramp3* gene to a negative influence of unknown nature. The pattern and intensity H3K27me3 suppressive marks were equivalent in mammary and non-mammary tissues and there was no change in mammary tissue upon loss of CTCF site E (Figure 9).

12. Page 4. “Experimental evidence that TAD boundaries and chromatin loops anchored by CTCF”. If a TAD boundary has a CTCF site, it because this site is an anchor for a loop. So the two are the same.

Response

We agree and corrected this.

13. Page 5. “The mammary-specific *Wap* super-enhancer activates the two juxtaposed promoters, but has little or no capacity to effectively reach over its native target gene to activate the downstream gene, even upon deletion of separating CTCF sites”. Please see concern expressed in #8 above about the possible role of additional CTCF sites within the *Wap* gene. I’m not sure this conclusion is supported by the data given the high occupancy of one of these CTCF sites in the DeltaAB mutant.

Response

We agree that additional CTCF sites in the body of the *Wap* gene could block the *Wap* super-enhancer. We addressed this in the discussion. In the future we contemplate an experiment taking out *Wap* sequences between the super-enhancer and *Tbrg4*.

14. Page 5. “Our finding that an anchor CTCF site serves a dual purpose, muffling a superenhancer and coincides with cell-specific regulatory features, suggests a more complex biological role for these elements”. It’s not clear what the authors mean by this. I don’t think that any of the results in the manuscript imply a more complex role for CTCF sites.

Response

We did not imply that CTCF itself has different functions. However, our data clearly demonstrate that loss of this specific CTCF site results in the reduction of *Ramp3* expression in several tissues analyzed. In contrast, increased expression was only observed in differentiated mammary tissue. There are two alternative explanations that were already discussed under point #11.

REVIEWERS' COMMENTS:

Reviewer #1 (Remarks to the Author):

The revised version of this manuscript is much improved. The authors have provided satisfactory answers to most of the concerns raised before, and the new data provided strengthen the conclusions. Although this manuscript remains limited by its focus on a single super-enhancer, the results provided do show "more complex and cell-specific functions of CTCF sites" than previously reported, which is therefore of interest for the scientific community interested in broad issues of gene regulation.

A few minor concerns remain:

- Lines 156-158 and line 201: It is not clear how the Authors define ESCs and T cells "cell types that respond to cytokines" and why is this relevant. The sentence "suggesting a possible role in cytokine-regulated gene expression" is not supported by data and should be removed.
- Several data points (trivial to acquire or add) are missing from the figures, and should be added for completeness:
 - Ramp3 gene in Fig 4d
 - Tbrg4 gene in Fig 4d-e
 - Delta CDE sample in Fig 6b
- Several figures lack labels on the y axis: 2b,c bottom panels, 3c, 4c,d,e, 6b, 7a,b 8c,d
- Several figure legends are stating results rather than explaining the figure itself. This makes the figures very difficult to understand.
- Figure 6: The RACE results should be presented. Does the black arrow represent RACE derived data? Do the black bars represent single reads in the RNAseq? it seems that most of the expression is around the second pol2 peak and away from S3. Can the authors clarify?

Reviewer #2 (Remarks to the Author):

I have read through the revised version of the manuscript. The authors have made a commendable effort to address all the issues raised by the reviewers. The only exception are a few laborious experiments that would take a long time to perform. I tend to agree with the authors that these additional experiments fall out of the main theme of the manuscript. I think this is a very nice piece of work on a very timely and interesting topic. My humble opinion is that the manuscript is appropriate for publication in Nature Communications in its current form.

Reviewer #1

1) Lines 156-158 and line 201: It is not clear how the Authors define ESCs and T cells "cell types that respond to cytokines" and why is this relevant. The sentence "suggesting a possible role in cytokine-regulated gene expression" is not supported by data and should be removed.

We changed those sentences.

2) Several data points (trivial to acquire or add) are missing from the figures, and should be added for completeness:

- Ramp3 gene in Fig 4d
- Tbrg4 gene in Fig 4d-e
- Delta CDE sample in Fig 6b

We have now added these data to the respective figures.

3) Several figures lack labels on the y axis: 2b,c bottom panels, 3c, 4c,d,e, 6b, 7a,b 8c,d

We have added the missing labels.

4) Several figure legends are stating results rather than explaining the figure itself. This makes the figures very difficult to understand.

We have now revised the figure legends.

5) Figure 6: The RACE results should be presented. Does the black arrow represent RACE derived data? Do the black bars represent single reads in the RNAseq? it seems that most of the expression is around the second pol2 peak and away from S3. Can the authors clarify?

The RACE result is now shown in Supplementary Note 2.

Regarding the figure: Yes, the black arrow represents the RACE data and the black bars are single reads from the RNA-seq. We are aware of the high expression at the second Pol II peak and we are currently analyzing this further.